# Deoxynivalenol Induces Local Inflammation and Lesions in Tissues at Doses Recommended by the EU

**DOI:** 10.3390/ijms25189790

**Published:** 2024-09-10

**Authors:** Alix Pierron, Luciana C. Balbo, Laura Soler, Philippe Pinton, Sylvie Puel, Joëlle Laffitte, Mickaël Albin, Ana-Paula F. R. Loureiro Bracarense, Maria A. Rodriguez, Isabelle P. Oswald

**Affiliations:** 1Toxalim (Research Centre in Food Toxicology), Université de Toulouse, INRAE, ENVT, INP-Purpan, UPS, 31027 Toulouse, France; alix.pierron@envt.fr (A.P.); laura.soler@inrae.fr (L.S.); philippe.pinton@inrae.fr (P.P.); sylvie.puel@inrae.fr (S.P.); jlaffitte@posteo.net (J.L.); mickael.albin@inrae.fr (M.A.); 2Laboratory of Animal Pathology, Universidade Estadual de Londrina, Londrina 86057-970, Brazil; luciana.cbalbo@uel.br (L.C.B.); anapaula@uel.br (A.-P.F.R.L.B.); 3Olmix, 56580 Bréhan, France; mrodriguez@olmix.com

**Keywords:** mycotoxins, toxicity, DON, histology, immunity-related genes

## Abstract

The mycotoxin deoxynivalenol (DON) is frequently present in cereals at low levels, resulting in its occurrence in food and feed. DON has been proven to alter the immune response and induce inflammation in all species, with pigs exhibiting heightened sensitivity and exposure. However, no study has yet evaluated the effects of exposure to DON at the recommended levels in pig feed. In two separate trials, piglets were subjected to control feed or feed contaminated with a low level of purified DON (0.83 mg/kg feed in trial 1 and 0.85 mg/kg feed in trial 2) for either three weeks (trial 1) or two weeks (trial 2). Additionally, a group of animals exposed to 2.85 mg/kg feed of DON was included as a positive control in Trial 1. The impact of DON on porcine tissues (intestine, liver, and spleen) was evaluated through histological and qPCR analyses of immune-related genes. Additionally, biochemical analyses and acute-phase proteins were examined in plasma samples. Lesions were identified in the intestine (jejunum and ileum), the liver, and the spleen of pigs receiving diets contaminated with low and high concentrations of DON. The low level of DON also resulted in impaired expression of genes associated with intestinal barrier integrity, intestinal immune responses, and liver function. In conclusion, the results of the two trials demonstrate the impact of DON exposure even at doses below the recommended level of 0.9 mg/kg feed set by the European Union. This suggests that the current recommended level should be reconsidered to ensure the optimal health and well-being of pigs.

## 1. Introduction

Mycotoxins are toxic secondary metabolites that are naturally produced by different types of fungi and contaminate plant-derived products [1]. Recent worldwide surveys on mycotoxin contamination have pointed out that 90% of food crops are contaminated with mycotoxins [2].

One of the most common mycotoxins in the world is deoxynivalenol (DON), produced by several Fusarium species such as *F. graminearum* and *F. culmorum*. DON is very stable and resistant to most feed manufacturing processes and so can be found from low (<1 mg/kg) to high (5–20 mg/kg) concentrations in cereals and at all stages of the feed chain [3]. Because of its occurrence and its acute and chronic toxicity, DON can represent a serious problem for both human and animal health [3,4,5,6]. The acute effects of DON include abdominal discomfort, diarrhea, vomiting, leukocytosis, gastrointestinal bleeding, and circulatory shock, while chronic exposure results in decreased feed intake, impaired growth, impaired intestinal barrier function, and gastrointestinal inflammation [7,8,9,10,11].

At the molecular level, DON is capable of binding to the 60S subunit of the ribosome, thereby disrupting subsequent protein synthesis and leading to the activation of mitogen-activated protein kinases (MAPKs) [12,13]. This results in a range of phenomena, including immunomodulation and local inflammation [10,13,14], which have been observed in a range of in vitro models and animal species [14,15,16,17,18].

Among all animals, pigs exhibit the greatest sensitivity to DON, even at relatively low doses [9]. Indeed, DON is rapidly absorbed and bioavailable, and as a monogastric, pigs are not well equipped to metabolize it effectively [19].

Few studies assessed in vivo the toxicity, both on immunity and locally on tissues, of DON at low levels, and none below the limit recommended by the European Union (0.9 mg/kg; [20,21]).

The objective of this study was to assess the effects on local inflammation and tissue lesions in piglets exposed to diets contaminated by concentrations of DON below the current European Union’s recommendation.

## 2. Results

### 2.1. Low Levels of DON Have an Impact on the Intestine

The effects of exposure to a low and high dose of DON were investigated in several tissues starting with the intestine, which is the first organ exposed after mycotoxin ingestion. In both trials (trial 1 lasting 3 weeks and trial 2 lasting 2 weeks), the lesions observed in the small intestine were mild to moderate in all animals exposed to DON. The main histological changes observed in the intestine of animals treated with DON were villi atrophy and fusion, with flattening of enterocytes and denuded villi. In both the jejunum (Figure 1) and the ileum (Figure 2), the lesion scores observed in animals receiving the DON-contaminated diet were significantly higher than those observed in animals fed the control diet. Interestingly, the lesion score in the jejunum was almost the same for low- and high-DON groups.

To assess the transcriptional modifications that can occur in the intestine, gene modulation was analyzed by RT q-PCR in trial 2 (Figure 1). In the jejunum of animals treated with a low dose of DON, a trend of increased expression of the nuclear factor kappa B subunit 1 (NFKB1), the insulin-like growth factor (IGF-1), and the mucin producer gene (MUC2) was observed in comparison to the control animals (Figure 1). In parallel, a trend of decreased expression of cytokines and genes implicated in the immune response (IL-1β, IL-1α, IFN-γ, TGF-β1) was observed in animals receiving feed contaminated with a low dose of DON compared to the control animals.

### 2.2. Low Levels of DON Have an Impact on the Liver

The effects of exposure to a diet containing a low dose of DON on the liver were then analyzed. As observed in the intestine, the hepatic lesions in the animals from the two trials were mild to moderate. Ingestion of DON caused the disorganization of hepatocytes and cytoplasmic vacuolization and megalocytosis of the same cells (Figure 3). A significant dose-dependent increase in the liver lesion score was observed in animals fed the DON-contaminated diet.

Concerning the modulation of gene expression in the liver, a significant decrease in the gene expression of ABCA1, implicated in the cholesterol transport, was seen in the low-DON group (Figure 3). The other genes implicated in the immune response (IL-1α) and in satiety (CCK) show a trend of decreased expression in the low-DON group. In contrast, the anti-inflammatory gene IL-10 and genes implicated in oxidative stress (SOD1, SOD2) show a trend of increased expression.

The effect of ingestion of a diet contaminated with low and high levels of DON on plasmatic biochemical analytes, as biomarkers of organ function and lesion, was also investigated (Figure 4). Overall, LDL, HDL, total cholesterol, AST, glucose PAP, and albumin were slightly decreased in the serum of animals receiving the DON-contaminated diets, but stayed within the physiological range. 

### 2.3. Low Levels of DON Have an Impact on the Spleen and Inflammatory Proteins

The last organ examined was the spleen. The lesions observed were mild to moderate. Lymphoid depletion and apoptosis of lymphocytes were the most common alterations observed in the spleen (Figure 5). A non-significant increase in the lesion score was observed in the spleen of animals receiving the DON-contaminated diets compared to the control animals.

Several genes of the immune and inflammatory response showed a trend of decreased expression (*p* < 0.1) in the presence of low doses of DON, Il-10, CXCL8, IFN-γ, and TNF-α (Figure 5). The other genes implicated in inflammation (IL-12, NFKB1), oxidative stress/transcription factor (SOD1, SOD2, p53), and immune defense (Casp3, Casp9) also tended to decrease in the presence of low doses of DON, but the trend was not significant.

The pig-MAP (rapid response), an acute-phase protein biomarker of inflammation, was measured in the plasma of all of the pigs (Figure 6). A slight increase in the pig-MAP level was observed in the plasma of animals fed diets contaminated with low levels of DON, while a significant decrease was observed in animals fed the highly contaminated diet.

## 3. Discussion

The toxicity of DON is well documented [3,4,5,6,7,10,22], and a number of in vivo experiments have been conducted over the years, particularly in pigs [8,23], but most did not assess the effects of this toxin at the dose recommended by the EU. This paper aims to address this gap.

In the present experiment, piglets were exposed to low doses of DON for 2 or 3 weeks. During the exposure period, the animals treated with DON in the two trials exhibited lower weight and weight gain, which is consistent with the observed trend of decreased feed intake. Few papers also show the same effects on growth performances at low doses. Prelusky shows reduced body weight gain and feed intake at 0.75 mg/kg feed (contaminated corn) during the first week [24]. Reduced feed intake and body weight gain were also seen at 0.38 and 0.75 mg/kg feed with pigs receiving DON-contaminated wheat for 21 days [25]. Alizadeh shows reduced body weight gain at 0.9 mg/kg feed of pure DON for 10 days; however, the animals received a bolus with a high concentration of DON (0.28 mg/kg body weight) 2 h before the procedure of euthanasia [26].

The histopathological damages observed in the intestine, the liver, and the spleen were comparable to the effects previously documented in the literature for DON at higher concentrations. Alterations in the intestinal morphology, with reduced villi height and fusion of the villi, have already been observed in pigs exposed to 0.9 to 7 mg DON/kg feed [26,27,28,29]. Similarly, liver changes, including hepatocyte disorganization and vacuolization and megalocytosis of liver cells, have been previously described following exposure to feed contaminated with 12 µg/kg BW per day or to 3 mg DON/kg feed [29,30]. Splenic lymphoid depletion and cell apoptosis have also been observed in pigs fed DON at 3 and 7 mg/kg BW [27,28]. Alizadeh also shows histomorphological alterations and modulation of the expression of mRNA encoding several tight junction proteins or inflammatory cytokines in animals exposed to 0.9 mg DON/Kg feed for 10 days and receiving a high dose of DON by bolus before euthanasia [26]. These histological lesions indicate that the low dose of DON is harmful to the animals, as recently pointed out by EFSA in broiler chickens and turkeys [31].

Inflammation was assessed by measuring the acute-phase proteins (pig-MAP), which have been demonstrated to be a reliable biomarker of inflammation in subclinical disease and a potential biomarker for pig health [32,33,34]. Our findings demonstrate that exposure to a low dose of DON tends to increase the pig-MAP level; the slight increase could be due to the time frame. In contrast, a high dose leads to a decrease, which may illustrate the known anti-inflammatory effect of DON at higher doses and could be due to the disruption of protein synthesis and the increased consumption of those proteins, as already proposed [35].

To go further, gene expression was investigated in the jejunum, liver, and spleen. Our findings demonstrate that low doses of DON modulate inflammatory cytokines. The inflammatory profile in the jejunum is evidenced by the decrease in TGF-β. The decrease in IFN-γ suggests that inflammation is mediated by antigen-presenting cells (APCs) and TH17 cells rather than by TH1 cells. DON induces a ribotoxic stress and targets NFkB, leading to the modulation of several genes, with a significant impact on immunity, inflammation, and oxidative stress [36,37]. In the liver, this study shows that exposure to a low dose of DON caused an anti-inflammatory response, with an increase in IL-10 and a decrease in IL-1α gene expression. In pigs, 21-day exposure to feed contaminated with a high dose of DON (3.1 mg/kg feed) resulted, by contrast, in an increased expression of several inflammatory cytokines [38]. Similarly, exposure of porcine liver slices or human hepatic cells induced histological damages and increased expression of genes involved in apoptosis and inflammation [18]. Additionally, hematological analysis indicated no significant modulation of white blood cells. These results clearly demonstrate the adverse effects of 0.9 mg DON/Kg feed, the value retained by the EC Commission and commonly used by all feed manufacturers.

It is important to consider that food and feed can be co-contaminated with a multitude of mycotoxins that can interact synergistically [39,40]. Additionally, animals can be exposed to both mycotoxins and pathogens, and numerous studies have indicated that DON enhances susceptibility to infectious diseases [16,41,42].

In conclusion, the findings of this study indicate that the EU regulatory level of 0.9 mg DON/kg feed induces lesions in the intestine, liver, and spleen, and the modulation of the inflammatory response. This suggests that the current recommended level should be reconsidered in order to ensure the optimal health and well-being of pigs.

## 4. Materials and Methods

### 4.1. Experimental Design and Tissue and Blood Sampling

The two animal trials were carried out in accordance with the European Directive on the protection of animals used for scientific purposes (Directive 2010/63/EU). The procedures were validated by the Ethics Committee for Animal Experiments Toxcomethique n°86 (TOXCOM/255/API). Thirty 4-week-old weaned crossbred castrated male pigs were obtained locally (GAEC Calvignac, St Vincent d’Autejac, France). Prior to being allocated to the different groups (6 animals per group), the animals were acclimatized for 11–13 days in the animal facility of the Toxalim Laboratory (Toulouse, France). Then, the animals were either exposed for three weeks (trial 1) or two weeks (trial 2) to the different diets (non-contaminated, contaminated with 0.9 or 3 mg/kg feed DON, Appendix A).

The diets were formulated as described in Appendix A. They were prepared at the INRAE facilities in St Gilles (France) or at ABZ Diervoeding (Nijkerk, The Netherland) for trial 1 and 2, respectively. Purified DON purchased from Sigma-Aldrich (Sigma, St Quentin Fallavier, France) was diluted in the premix prior to its incorporation into the diet feed during manufacture. The targeted DON concentrations were 0.9 and 3 mg/kg feed, and the exact concentrations determined by LABOCEA (Ploufragan, France) were slightly lower (0.85 and 2.85 for trial 1, 0.83 mg/kg feed for trial 2) (Appendix A). This method analyzes 45 mycotoxins and metabolites, for which the level was below their limits of detection (Appendix A).

During the trials, at weekly intervals, blood samples were aseptically collected in heparin sodium tubes from the jugular vein (Vacutainer^®^, Becton-Dickinson, Franklin Lakes, NJ, USA). The blood was centrifugated at 2000× *g* for 10 min to obtain the plasma, which was stored at −20 °C for later analysis. At the end of the experiment, the animals were slaughtered by electronarcosis before exsanguination. The different tissues were collected (jejunum and ileum without Peyer’s patches, liver, and spleen) and stored at −80 °C for cytokine mRNA measurements or fixed in 10% buffered formalin for histopathological analysis.

### 4.2. Biochemical Analysis and Acute-Phase Protein Measurement

The different biochemical parameters of the plasma (HDL, LDL, cholesterol, albumin, alanine aminotransferase (ALT), aspartate aminotransferase (AST), total protein, glucose PAP) were measured at the Anexplo Platform in Toulouse (France).

The acute-phase protein dosage, pig-MAP, was realized using an ELISA kit (ACUVET ELISA pig-MAP, Acuvet Biotech, Zaragoza, Spain).

### 4.3. Histopathological Analysis

Tissues were stained with hematoxylin–eosin (HE) for histological evaluation. Histopathological scoring of jejunum, ileum, liver, and spleen tissues was performed as already described [27,28]. Morphometry of crypt and intestinal villi was assessed using a MOTIC Image Plus 2.0 ML^®^ image analysis system as already described [26,28].

### 4.4. Expression of mRNA Encoding Cytokines by Real-Time PCR

RNA from the different tissues (jejunum, ileum, liver, and spleen) was extracted as previously described [28]. The concentration and quality of the samples were analyzed; reverse transcription and real-time qPCR were performed as already described [43] using the primers indicated in Appendix A. LinRegPCR software (version 2016.0) and N0 values (starting concentrations) were used to analyze the data. The most appropriate combination of housekeeping genes for each organ was chosen among HMBS, B2M, and RPL32 by the software Normfinder_0953 and used to normalize the data. The expression level was expressed relative to the mean of the control group, as already described [28].

### 4.5. Statistical Analysis

For statistical analysis, a one-way ANOVA for histological and Q-PCR analyses and a two-way ANOVA for biochemistry and ELISA analysis were performed with a Bonferroni test as the post hoc test, with *p* < 0.05. All data that did not follow a normal distribution were log-transformed before the analysis. If the data still did not follow a normal distribution after the transformation, a Kruskal–Wallis non-parametric test, with Mann –Whitney as the post hoc test and *p* < 0.05, was performed.

All of the statistics were analyzed independently for the two trials.

## Figures and Tables

**Figure 1 ijms-25-09790-f001:**
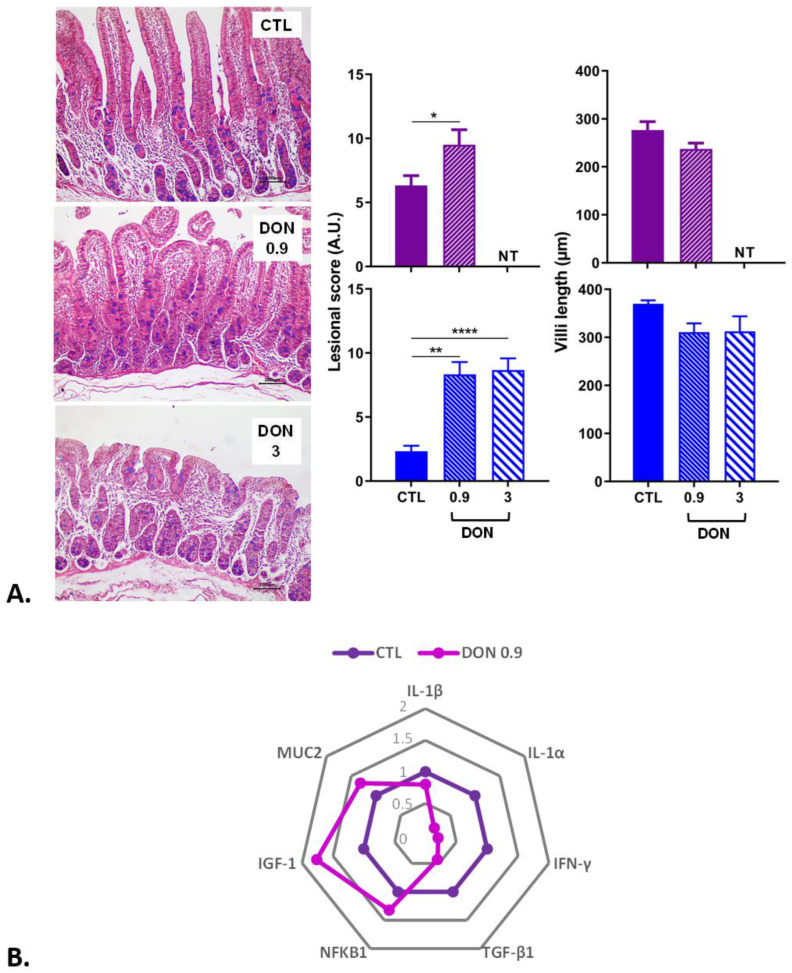
The effect of low and high doses of DON on jejunal histopathology and gene expression (n = 6 animals/group). (**A**) Histological analyses: two trials were performed (trial 1 in blue lasted three weeks; trial 2 in purple lasted two weeks). Lesional scores are indicated in arbitrary units (A.U.). * *p* < 0.05, ** *p* < 0.01, and **** *p* < 0.0001; NT: not tested for this trial. A representative image is also displayed for each treatment. Control, no significant histological change. DON 0.9 mg/kg feed, moderate villi atrophy. DON 3 mg/kg feed, severe villi atrophy and villi fusion. HE. Scale bar, 100 µm. (**B**) Gene expression levels. mRNA levels of inflammatory and intestinal integrity markers were measured in trial 2 by RT q-PCR. The control group is represented in dark purple and the DON group (0.9 mg/kg feed) in light purple. The results are expressed as the relative mRNA expression.

**Figure 2 ijms-25-09790-f002:**
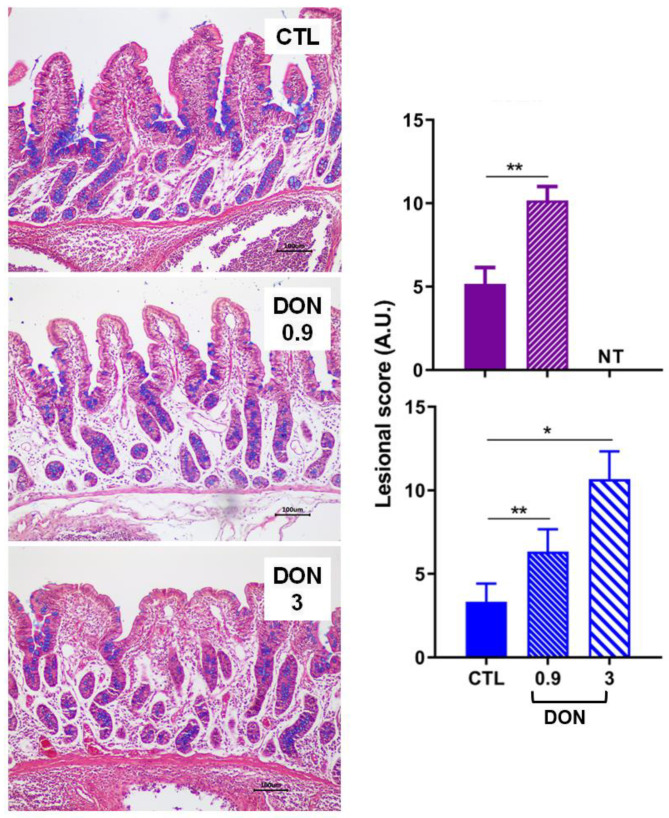
The effect of low and high doses of DON on ileal histopathology (n = 6 animals/group). Two trials were performed (trial 1 in blue lasted three weeks; trial 2 in purple lasted two weeks). Lesional scores are indicated in arbitrary units (A.U.). * *p* < 0.05, ** *p* < 0.01; NT: not tested for this trial. A representative image is also displayed for each treatment. Control, no significant histological change. DON 0.9 mg/kg feed, moderate to severe edema of lamina propria, lymphatic dilation, and mild villi atrophy. DON 3 mg/kg feed, moderate edema of lamina propria, and villi atrophy and fusion. HE. Scale bar, 100 µm.

**Figure 3 ijms-25-09790-f003:**
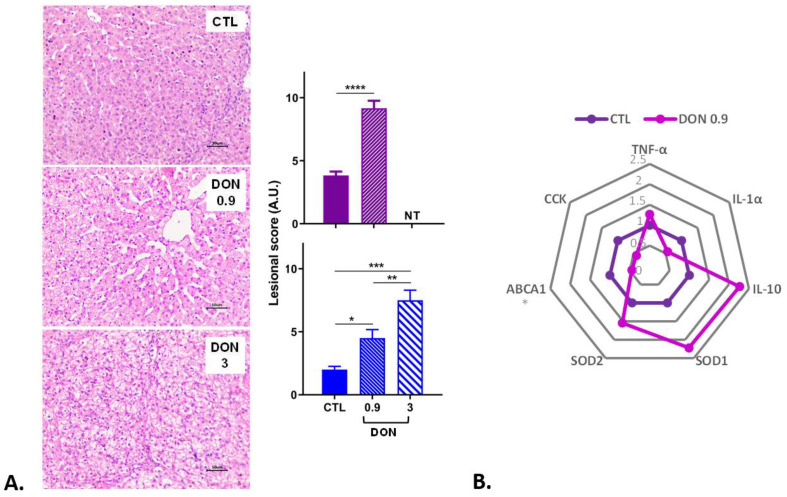
The effect of low and high doses of DON on liver histopathology and gene expression (n = 6 animals/group). (**A**) Histological analyses: two trials were performed (trial 1 in blue lasted three weeks; trial 2 in purple lasted two weeks). Lesional scores are indicated in arbitrary units (A.U.). * *p* < 0.05, ** *p* < 0.01, *** *p* < 0.001 and **** *p* < 0.0001; NT: not tested for this trial. A representative image is also displayed for each treatment. Control, no significant histological change. (**B**) DON 0.9 mg/kg feed, disorganization of hepatocyte trabeculae and mild cytoplasmic vacuolation of hepatocytes. DON 3 mg/kg feed, diffuse cytoplasmic vacuolation of hepatocytes. HE. Scale bar, 50 µm. B. Gene expression levels. mRNA levels of inflammatory, apoptosis, and oxidative stress markers were measured in trial 2 by RT q-PCR. The control group is represented in dark purple and the DON group (0.9 mg/kg feed) in light purple. The results are expressed as the relative mRNA expression; * *p* < 0.05.

**Figure 4 ijms-25-09790-f004:**
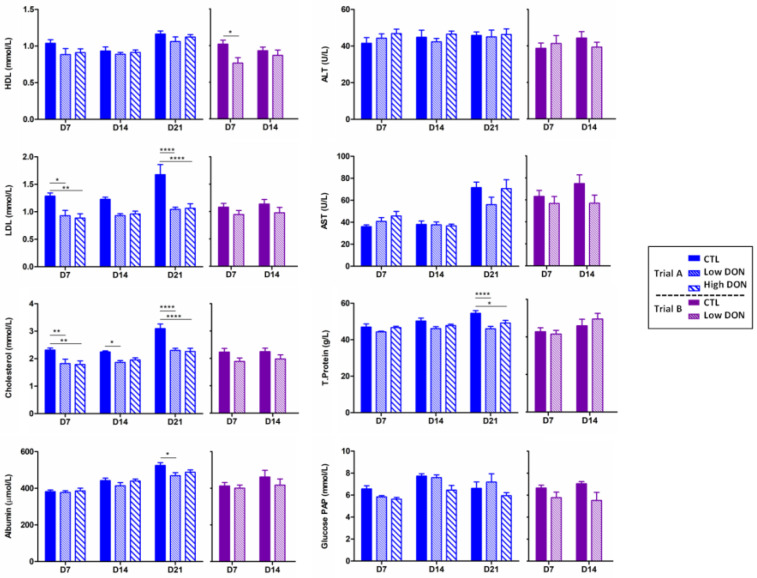
The effects of DON on biochemical parameters at low and high doses. HDL (high-density lipoprotein), ALT (alanine transaminase), LDL (low-density lipoprotein), AST (aspartate aminotransferase), total cholesterol, albumin, and glucose PAP were measured each week in the plasma of all piglets for the two trials. Trial 1 is depicted in dark blue, trial 2 in purple, and the control group with full color. DON 0.9 is represented with fine dashes and DON 0.3 with thick dashes. The data are expressed as the mean ± SEM for n = 6; * *p* < 0.05, ** *p* < 0.01 and **** *p* < 0.0001.

**Figure 5 ijms-25-09790-f005:**
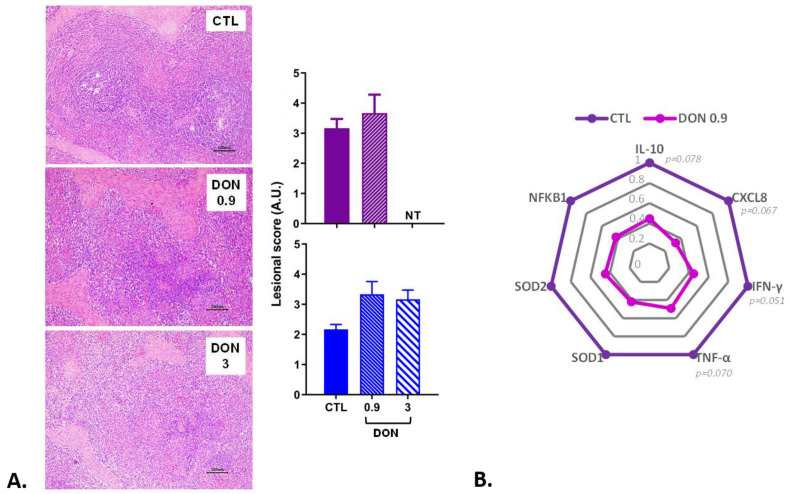
The effect of low and high doses of DON on splenic histopathology and gene expression (n = 6 animals/group). (**A**) Histological analyses: two trials were performed (trial 1 in blue lasted three weeks; trial 2 in purple lasted two weeks). Lesional scores are indicated in arbitrary units (A.U.). *p* < 0.05; NT: not tested for this trial. A representative image is also displayed for each treatment. Control, no significant histological change. DON 0.9 mg/kg feed, moderate lymphoid depletion. DON 3 mg/kg feed, diffuse lymphoid depletion. HE. Scale bar, 50 µm. (**B**) Gene expression levels. mRNA levels of inflammatory and oxidative stress markers were measured in trial 2 by RT q-PCR. The control group is represented in dark purple and the DON group (0.9 mg/kg feed) in light purple. The results are expressed as the relative mRNA expression; trend *p* < 0.1.

**Figure 6 ijms-25-09790-f006:**
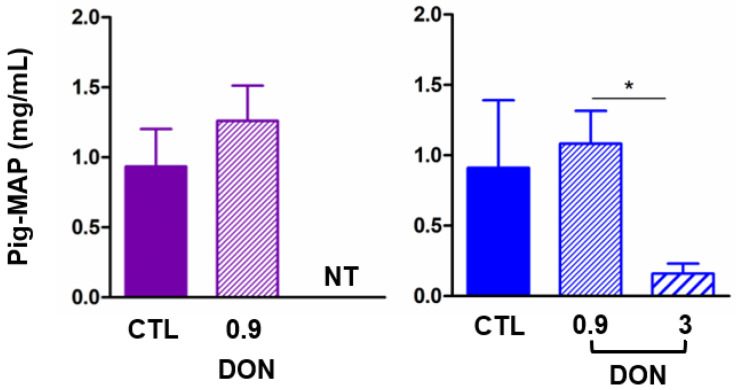
Analysis of effect of low DON levels on one acute-phase protein in plasma. Level of pig-MAP in pig plasma at D14 for two trials in pigs receiving 0, 0.9, or 3 mg/kg DON in feed. Data are expressed as mean ± SEM for n = 6; * *p* < 0.05. NT means not tested for this trial.

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
