# Peer review of "Deoxynivalenol Induces Local Inflammation and Lesions in Tissues at Doses Recommended by the EU"

_ijms, 2024, doi:10.3390/ijms25189790_

Round 1

Reviewer 1 Report

Comments and Suggestions for Authors

The authors of the MS entitled, the inflammatory mycotoxin, deoxynivalenol, induces local inflammation and lesions in tissues at doses below the EU recommended level” evaluated

the effect of exposure to deoxynivalenol (DON) at the various levels (recommended level included) in pig feed.

Briefly, in one of the trials that were performed, piglets were exposed to control feed that contained a low dose of purified DON (e.g. 0.83 mg/kg) where the recommended level is 0.9 mg/kg in Europa.  The expected conclusions, after several evaluations (e.g., different concentrations 0.83 mg/kg vs 0.85 mg/kg of pig feed for three vs two weeks) was that changes need to be made regarding the DON exposure  at low doses.

The data are interesting and could be helpful. However, the manuscript needs to be shortened and the text made easier to read. one example among many others:

“Figure 1.  Histopathological effects of DON at low and high dose on jejunum”

Perhaps it could read better as follows: Effect of low and high doses of DON……

Similar changes for the other figure should be made; also, there are too many figures and they are isolated from the “results”, which makes the MS more difficult to interpret.

The “Discussion” could be shortened by half, again for the reason stated above.

Comments on the Quality of English Language

/

Author Response

The authors of the MS entitled, the inflammatory mycotoxin, deoxynivalenol, induces local inflammation and lesions in tissues at doses below the EU recommended level” evaluated the effect of exposure to deoxynivalenol (DON) at the various levels (recommended level included) in pig feed.

Briefly, in one of the trials that were performed, piglets were exposed to control feed that contained a low dose of purified DON (e.g. 0.83 mg/kg) where the recommended level is 0.9 mg/kg in Europa.  The expected conclusions, after several evaluations (e.g., different concentrations 0.83 mg/kg vs 0.85 mg/kg of pig feed for three vs two weeks) was that changes need to be made regarding the DON exposure  at low doses.

The data are interesting and could be helpful. However, the manuscript needs to be shortened and the text made easier to read. one example among many others:

Comment 1:“Figure 1.  Histopathological effects of DON at low and high dose on jejunum”

Perhaps it could read better as follows: Effect of low and high doses of DON……

Response 1: This figure has been combined with Figure 3 and the title has been changed to Effect of low and high doses of DON on jejunal histopathology and cytokine gene expression.

Comment 2: Similar changes for the other figure should be made; also, there are too many figures and they are isolated from the “results”, which makes the MS more difficult to interpret.

Response 2: In the revised version the number of figures have been reduced from 9 to 6 and their titles have been changed.

Comment 3: The “Discussion” could be shortened by half, again for the reason stated above.

Response 3: The discussion is quite short (only 1117 words, less than 2 pages), to answer this comment we have reduced it by 1/3. 

Reviewer 2 Report

Comments and Suggestions for Authors

The manuscript describes local inflammation and tissue lesions at different doses in pigs. It is perfectly planned, elaborated and exposed, so its content is of great interest for researchers interested in the toxicity of DON, besides it can serve as a model for the study of the toxicity of other mycotoxins. Consequently, it deserves to be published.

However, some formal considerations can be made:

- Continuous reference is made in the text to addressing a study at lower doses than those recommended by the EU, but perhaps it is more reasonable to say at the same doses, given that 0.83 -0.85 ppm (doses used) is practically equal to 0.9 ppm (EU recommended dose). In fact, as stated in the experimental part, it was intended to introduce 0.9 ppm of DON in the feed and when analyzing the contaminated feed, 0.83-0.85 ppm was found. Moreover, the cited legislation has been recently updated with the COMMISSION REGULATION (EU) 2024/1022 of 8 April 2024, amending Regulation (EU) 2023/915 as regards maximum levels of deoxynivalenol in food. It should be mentioned in the introduction.

- In the first paragraph of the introduction, reference is made to 60-80% of the crops being contaminated by mycotoxins, which in the light of the latest advances in mycotoxin analysis seems a low percentage, more realistic would be to refer to 90%, so the paragraph should be reworked and perhaps eliminate references 1 and 3.

- In Figure 1 and subsequent figures, trial A and B are referred to, and images named A and B are shown, in order to facilitate the reader's understanding it would be more appropriate to make changes, for example to name trials 1 and 2.

- It would be important to provide some explanation for the fact that plasma proteins increase slightly at the lowest DON dose and decrease considerably at 3 ppm (lines 155-159 and 217-223).

The authors should consider changing the title of the article to "Deoxynivalenol induces local inflammation and lesions in pigs at doses recommended by the EU". With this, the final important message that the limits should be reconsidered remains equally valid.

As keywords, words other than those contained in the title of the article should be introduced, perhaps: "mycotoxins, toxicity, DON, histology, immunity-related genes, proteins".

Author Response

The manuscript describes local inflammation and tissue lesions at different doses in pigs. It is perfectly planned, elaborated and exposed, so its content is of great interest for researchers interested in the toxicity of DON, besides it can serve as a model for the study of the toxicity of other mycotoxins. Consequently, it deserves to be published.

However, some formal considerations can be made:

Comment 1: Continuous reference is made in the text to addressing a study at lower doses than those recommended by the EU, but perhaps it is more reasonable to say at the same doses, given that 0.83 -0.85 ppm (doses used) is practically equal to 0.9 ppm (EU recommended dose). In fact, as stated in the experimental part, it was intended to introduce 0.9 ppm of DON in the feed and when analyzing the contaminated feed, 0.83-0.85 ppm was found. Moreover, the cited legislation has been recently updated with the COMMISSION REGULATION (EU) 2024/1022 of 8 April 2024, amending Regulation (EU) 2023/915 as regards maximum levels of deoxynivalenol in food. It should be mentioned in the introduction.

Response 1: The expression below recommended level has been changed throughout the manuscript and the regulation has been updated

Comment 2: In the first paragraph of the introduction, reference is made to 60-80% of the crops being contaminated by mycotoxins, which in the light of the latest advances in mycotoxin analysis seems a low percentage, more realistic would be to refer to 90%, so the paragraph should be reworked and perhaps eliminate references 1 and 3.

Response 2: the changes have been made and the reference eliminated

Comment 3: In Figure 1 and subsequent figures, trial A and B are referred to, and images named A and B are shown, in order to facilitate the reader's understanding it would be more appropriate to make changes, for example to name trials 1 and 2.

Response 3: Done

Comment 4:It would be important to provide some explanation for the fact that plasma proteins increase slightly at the lowest DON dose and decrease considerably at 3 ppm (lines 155-159 and 217-223).

Response 4: The slight increase at low DON dose could be due to the timeframe, that we are not at the highest point and already in the stage of decrease.

Considering the decrease, its less recorded but was also observed on several acute phase proteins in an experiment on pigs at high dose of 4.59 mg DON/kg feed during 4 weeks (Dänicke et al. 2020). They made the hypothesis that the decrease seen could be due to the depression in synthesis of these proteins or an imbalance between synthesis and usage.

This discussion has been updated and a reference added (Lines 203-207).

Dänicke S, Bannert E, Tesch T, Kersten S, Frahm J, Bühler S, Sauerwein H, Görs S, Kahlert S, Rothkötter HJ, Metges CC, Kluess J. Oral exposure of pigs to the mycotoxin deoxynivalenol does not modulate the hepatic albumin synthesis during a LPS-induced acute-phase reaction. Innate Immun. 2020. 26:716-732. doi: 10.1177/1753425920937778.

Comment 5: The authors should consider changing the title of the article to "Deoxynivalenol induces local inflammation and lesions in pigs at doses recommended by the EU". With this, the final important message that the limits should be reconsidered remains equally valid.

Response 5: The title has been changed and replaced by the suggested one.